# A Theory of Decision Making Under Dynamic Context

**Michael Shvartsman**
Princeton Neuroscience Institute
Princeton University
Princeton, NJ, 08544
ms44@princeton.edu

**Vaibhav Srivastava**
Department of Mechanical and Aerospace Engineering
Princeton University
Princeton, NJ, 08544
vaibhavs@princeton.edu

**Jonathan D. Cohen**
Princeton Neuroscience Institute
Princeton University
Princeton, NJ, 08544
jdc@princeton.edu

## Abstract

The dynamics of simple decisions are well understood and modeled as a class of random walk models [e.g. 1–4]. However, most real-life decisions include a dynamically-changing influence of additional information we call context. In this work, we describe a computational theory of decision making under dynamically shifting context. We show how the model generalizes the dominant existing model of fixed-context decision making [2] and can be built up from a weighted combination of fixed-context decisions evolving simultaneously. We also show how the model generalizes recent work on the control of attention in the Flanker task [5]. Finally, we show how the model recovers qualitative data patterns in another task of longstanding psychological interest, the AX Continuous Performance Test [6], using the same model parameters.

## 1 Introduction

In the late 1940s, Wald and colleagues developed a sequential test called the sequential probability ratio test (SPRT; [7]). This test accumulates evidence in favor of one of two simple hypotheses until a log likelihood threshold is crossed and one hypothesis is selected, forming a random walk to a decision bound. This test was quickly applied as a model of human decision making behavior both in its discrete form [e.g. 1] and in a continuous realization as biased Wiener process (the Diffusion Decision Model or DDM; [2]). This work has seen a recent revival due to evidence of neurons that appear to reflect ramping behavior consistent with evidence accumulation [e.g. 8], cortical circuits implementing a decision process similar to the SPRT in the basal ganglia in rats [9], and the finding correlations between DDM parameters and activity in EEG [10] and fMRI [11].

Bolstered by this revival, a number of groups investigated extension models. Some of these models tackle complex hypothesis spaces [e.g. 12], or greater biological realism [e.g. 13]. Others focus on relaxing stationarity assumptions about the task setting, whether by investigating multi-stimulus integration [5], deadlines [14], or different evidence distribution by trial [15].

We engage with the latter literature by providing a theory of multi-alternative decision making under dynamically changing context. We define context simply as additional information that may bear upon a decision, whether from perception or memory. Such a theory is important because even simple tasks that use apparently-fixed contexts such as prior biases may require inference on the

context itself before it can bear on the decision. The focus on dynamics is what distinguishes our work from efforts on context-dependent changes in preferences [e.g. 16] and internal context updating [e.g. 17]. The admission of evidence from memory distinguishes it from work on multisensory integration [e.g. 18].

We illustrate such decisions with an example: consider seeing someone that looks like a friend (a target stimulus), and a decision: to greet or not greet this person. A context can be external (e.g. a concert hall) or internal (e.g. the memory that the friend went on vacation, and therefore this person is likely a lookalike). The context can strictly constrain the decision (e.g. greeting a friend in the street vs. the middle of a film), or only bias it (guessing whether this is a friend or lookalike after retrieving the memory of them on vacation). Regardless, context affects the decision, and we assume it needs to be inferred, either before or alongside the greeting decision itself. We aim to build a normative theory of this context processing component of decision making. We show that our theory generalizes the discrete-time context-free SPRT (and therefore a Wiener process DDM in continuous time) and how context-dependent decisions can be optimally built up from a dynamically weighted combination of context-free decisions.

Our theory is general enough to consider a range of existing empirical paradigms in the literature, including the Stroop, Flanker, Simon, and the AX-CPT [6, 19–21]. We choose to mention these in particular because they reside on the bounds of the task space our theory considers on two different dimensions, and describe a discretization of task space on those dimensions that accommodates those existing paradigms. We show that in spite of the framework's generality, it can provide well-behaved zero-parameter predictions across qualitatively different tasks. We do this by using our framework to derive a notational variant of an existing Flanker model [5], and using parameter values from this previous model to simultaneously generate qualitatively accurate predictions in both the flanker and AX-CPT paradigms. That is, our theory generates plausible behavior *in qualitatively different tasks, using the same parameters*.

## 2  The theoretical framework

We assume that dynamic context decision making, like fixed context decision making, can be understood as a sequential Bayesian inference process. Our theory therefore uses sequentially drawn samples from external input and/or internal memory to compute the joint posterior probability over the identity of the true *context* and decision *target* over time. It maps from this joint probability to a response probability using a fixed response mapping, and uses a fixed threshold rule defined over the response probability to stop sampling and respond. We make a distinction between our *theory* of decision making and individual *task models* that can be derived from the theory by picking points in task space that the theory accommodates.

Formally, we assume the decider conditions a decision based on its best estimate of two pieces of information: some unknown true *context* taking on one of the values $\{c_i\}_{i=0}^n$, and some unknown true *target* taking on one of the values $\{g_j\}_{j=0}^m$. This intentionally abstracts from richer views of context (e.g. ones which assume that the context is qualitatively different from the target, or that the relevant context to sample from is unknown). We denote by $C, G$ random variables representing the possible draws of context and target, and $r(\cdot)$ a deterministic function from the distribution $P(C, G)$ to a distribution over responses. We define an abstract *context sensor* and *target sensor* selectively tuned to context or target information, such that $e^C$ is a discrete piece of evidence drawn from the context sensor, and $e^G$ one drawn from the target sensor. The goal of the decider is to average over the noise in the sensors to estimate the pair $(C, G)$ sufficiently to determine the correct response, and we assume that this inference is done optimally using Bayes' rule.

We denote by $t_c^{on}$ the time at which the context appears and $t_c^{off} \geq t_c^{on}$ the time at which it disappears, and likewise $t_g^{on} \leq t_g^{off}$ the time at which the target appears and disappears. We also restrict these times such that $t_c^{on} \leq t_g^{on}$; this is the primary distinction between context and target, which can otherwise be two arbitrary stimuli. The onsets and offsets define one dimension in a continuous space of tasks over which our theory can make predictions.

The form of $r(\cdot)$ defines a second dimension in the space of possible tasks where our theory makes predictions. We use a suboptimal but simple threshold heuristic for the decision rule: when the a

posteriori probability of any response crosses some adaptively set threshold, sampling ends and the response is made in favor of that response.

For the purposes of this paper, we restrict ourselves to two extremes on both of these dimensions. For stimulus onset and offset times, we consider one setting where the context and target appear and disappear together (perfect overlap, i.e. $t_c^{on} = t_g^{on}$ and $t_c^{off} = t_g^{off}$), and one where the target appears some time after the context disappears (no overlap, i.e. $t_c^{off} \leq t_g^{on}$). We label the former the *external context model*, because the contextual information is immediately available, and the latter the *internal context model*, because the information must be previously encoded and maintained. The external context model is like the ongoing film context from the introduction, and the internal context is like knowing that the friend is on vacation.

For the response mapping function $r(\cdot)$ we consider one setting where the response is solely conditioned on the perceptual target (*context-independent* response) and one where the response is is conditioned jointly on the context-target pair (*context-dependent* response). The context-dependent response is like choosing to greet or not greet the friend at the movie theater, and the context-independent one is like choosing to greet or not greet the friend on the street.

In the lab, classic tasks like the Stroop, Flanker, and Simon [19–21] fall into the taxonomy as external-context tasks with a context-independent response, because the response is solely conditioned on the perceptual target. On the other side of both dimensions are tasks like the N-back task and the AX Continuous Performance Test [6]. In our consideration of these tasks, we restrict our attention to the case where there are only two possible context and target hypotheses. The sequential inference procedure we use can be performed for other numbers of potentially-dependent hypotheses and responses, though the analysis we show later in the paper relies on the $n = m = 2$ assumption and on indepednence between the two sensors.

## 3 External context update

First we describe the inference procedure in the case of perfect overlap of context and target. At the current timestep $\tau$, the decider has available evidence samples from both the context and the target ($e^C$ and $e^G$) and uses Bayes' rule to compute the posterior probability $P(C, G \mid e^C, e^G)$:

$$P_\tau(C = c, G = g \mid e^C, e^G) \propto P(e^C, e^G \mid C = c, G = g) P_{\tau-1}(C = c, G = g) \tag{1}$$

The first term is the likelihood of the evidence given the joint context-target hypothesis, and the second term is the prior, which we take to be the posterior from the previous time step. We use the flanker task as a concrete example. In this task, participants are shown a central target (e.g. an S or an H) surrounded on both sides by distractors ('flankers', more S or H stimuli) that are either congruent or incongruent with it. Participants are told to respond to the target only but show a number of indications of influence of the distractor, most notably an early period of below-chance performance and a slowdown or reduced accuracy with incongruent relative to congruent flankers [20]. We label the two possible target identities $\{g_0 = S, g_1 = H\}$ and the possible flanker identities $\{c_0 = S\_S, c_1 = H\_H\}$ with the underscore representing the position of the target. This gives us the two congruent possibilities $\{[C = c_0, G = g_0], [C = c_1, G = g_1]\}$ or [SSS,HHH] and the two incongruent possibilities $\{[C = c_0, G = g_1], [C = c_1, G = g_0]\}$ or [SHS,HSH]. The response mapping function marginalize over context identities at each timestep:

$$r(P(C, G)) = \begin{cases} r_0 & \text{with probability } \sum_c P(C = c, G = g_0) \\ r_1 & \text{with probability } \sum_c P(C = c, G = g_1) \end{cases} \tag{2}$$

The higher of the two response probabilities is compared to a threshold $\theta$ and when this threshold is crossed, the model responds. What remains is to define the prior $P_0(C, G)$ and the likelihood function $P(e^C, e^G | C, G)$ or its inverse, the sample generation function. For sample generation, we assume that the context and target are represented as two Gaussian distributions:

$$e^C \sim \mathcal{N}(\mu_c + \alpha_\mu \mu_g, \sigma_c^2 + \alpha_\sigma \sigma_g^2) \tag{3}$$

$$e^G \sim \mathcal{N}(\mu_g + \alpha_\mu \mu_c, \sigma_g^2 + \alpha_\sigma \sigma_c^2) \tag{4}$$

Here $\mu_c$ and $\mu_g$ are baseline means for the distributions of context and target, $\sigma_c^2$ and $\sigma_g^2$ are their variances, and the $\alpha$ scaling factors mix them, potentially reflecting perceptual overlap in the sensors. This formulation is a notational variant of an earlier flanker model [5], but we are able to derive it by describing the task in our formalism (we describe the exact mapping in the supplementary material). Moreover, we later show how this notational equivalence lets us reproduce both Yu and colleagues' results and data patterns in another task, using the same parameter settings.

# 4 Comparison to a constant-drift model

We now write the model in terms of a likelihood ratio test to facilitate comparison to Wald's SPRT and Wiener diffusion models. This is complementary to an earlier approach performing dynamical analysis on the problem in probability space [22]. First we write the likelihood ratio $Z$ of the full response posteriors for the two responses. Since the likelihood ratio and the max a posteriori probability are monotonically related, thresholding on $Z$ maps onto the threshold over the probability of the most probable response we described above.

$$Z = \frac{p(r(P(C,G)) = r_0 | e^C, e^G)}{p(r(P(C,G)) = r_1 | e^C, e^G)} \tag{5}$$

$$= \frac{\left( P(e^C, e^G \mid C = c_0, G = g_0) P_{\tau-1}(C = c_0, G = g_0) + P(e^C, e^G \mid C = c_1, G = g_0) P_{\tau-1}(C = c_1, G = g_0) \right)}{\left( P(e^C, e^G \mid C = c_0, G = g_1) P_{\tau-1}(C = c_0, G = g_1) + P(e^C, e^G \mid C = c_1, G = g_1) P_{\tau-1}(C = c_1, G = g_1) \right)} \tag{6}$$

For this analysis we assume that context and target samples are drawn independently from each other, i.e. that $\alpha_\mu = \alpha_\sigma = 0$ and therefore that $P(e^C, e^G \mid C, G) = P(e^C \mid C)P(e^G \mid T)$. We also index the evidence samples by time to remove the prior terms $P_{\tau-1}(\cdot)$, and introduce the notation $l_t(t_x) = P(e_t^G \mid G = g_x)$ and $l_t(c_x) = P(e_t^C \mid C = c_x)$ for the likelihoods, with $x \in \{0, 1\}$ indexing stimuli and $t \in \{t_{c^{on}} = t_{g^{on}} \ldots \tau\}$ indexing evidence samples over time. Now we can rewrite:

$$Z^\tau = \frac{P_0(C = c_0, G = g_0) \prod_t l_t(c_0)l_t(g_0) + P_0(C = c_1, G = g_0) \prod_t l_t(c_1)l_t(g_0)}{P_0(C = c_0, G = g_1) \prod_t l_t(c_0)l_t(g_1) + P_0(C = c_1, G = g_1) \prod_t l_t(c_1)l_t(g_1)} \tag{7}$$

$$= \frac{P_0(C = c_0)P(G = g_0 \mid C = c_0) \prod_t l_t(c_0)l_t(g_0) + P_0(C = c_1)P(G = g_0 \mid C = c_1) \prod_t l_t(c_1)l_t(g_0)}{P_0(C = c_0)P(G = g_1 \mid C = c_0) \prod_t l_t(c_0)l_t(g_1) + P_0(C = c_1)P(G = g_1 \mid C = c_1) \prod_t l_t(c_1)l_t(g_1)} \tag{8}$$

Divide both the numerator and the denominator by $\prod_t l_t(c_1)$:

$$Z^\tau = \frac{P_0(C = c_0)P(G = g_0 \mid C = c_0) \prod_t \frac{l_t(c_0)}{l_t(c_1)} l_t(g_0) + P_0(C = c_1)P(G = g_0 \mid C = c_1) \prod_t l_t(g_0)}{P_0(C = c_0)P(G = g_1 \mid C = c_0) \prod_t \frac{l_t(c_0)}{l_t(c_1)} l_t(g_1) + P_0(C = c_1)P(G = g_1 \mid C = c_1) \prod_t l_t(g_1)} \tag{9}$$

Separate out the target likelihood product and take logs:

$$\log Z^\tau = \sum_{t=1}^\tau \log \frac{l_t(g_0)}{l_t(g_1)} + \log \frac{P(G = g_0 \mid C = c_0) \frac{P_0(C=c_0)}{P_0(C=c_1)} \prod_t \frac{l_t(c_0)}{l_t(c_1)} + P(G = g_0 \mid C = c_1)}{P(G = g_1 \mid C = c_0) \frac{P_0(C=c_0)}{P_0(C=c_1)} \prod_t \frac{l_t(c_0)}{l_t(c_1)} + P(G = g_1 \mid C = c_1)} \tag{10}$$

Now, the first term is the Wald's sequential probability ratio test [7] with $z_g^\tau = \sum_t \log \frac{l_t(g_0)}{l_t(g_1)}$. In the continuum limit, it is equal to a Wiener diffusion process $dz_g = a_g dt + b_g dW$ with $a_g = \mathbb{E}[\log \frac{l(g_0)}{l(g_1)}]$ and $b_g^2 = \text{Var}[\log \frac{l(g_0)}{l(g_1)}]$ [1, 4]. We can relabel the SPRT for the target $z_g^\tau = \sum_t \log \frac{l_t(g_0)}{l_t(g_1)}$ and do the same for the context drift that appears on both numerator and denominator of the final term: $z_\tau^c = \sum_t \log \frac{l_t(c_0)}{l_t(c_1)}$ and $z_c^0 = \log \frac{P_0(C=c_0)}{P_0(C=c_1)}$. Then the expression is as follows:

$$\log Z^\tau = z_g^\tau + \log \frac{P(G = g_0 \mid C = c_0)e^{z_c^0 + z_c^\tau} + P(G = g_0 \mid C = c_1)}{P(G = g_1 \mid C = c_0)e^{z_c^0 + z_c^\tau} + P(G = g_1 \mid C = c_1)} \tag{11}$$

$\log Z^\tau$ in equation (11) comprises two terms. The first is the unbiased SPRT statistic, while the second is a nonlinear function of the SPRT statistic for the decision on the context. The nonlinear term plays the role of bias in the SPRT for decision on target. This rational dynamic prior bias is an advance over previous heuristic approaches to dynamic biases [e.g. 23].

Several limits of (11) are of interest: if the context and the target are independent, then the second term reduces to $\log\left( \frac{P(G=g_0)}{P(G=g_1)} \right)$, and (11) reduces to the biased SPRT for the target. If each target is equally likely given a context, then the nonlinear term in (11) reduces to zero and (11) reduces to the SPRT for the target. If each context deterministically determines a different target, then any piece of evidence on the context is equally informative about the target. Accordingly, (11) reduces to the sum of statistic for context and target, i.e., $z_g^\tau \pm (z_c^\tau + z_c^0)$. If the magnitude of drift rate for the context is much higher than the magnitude of drift rate for the target, or the magnitude of the bias $z_0^c$ is high, then the nonlinear term saturates at a faster timescale compared to the decision time. In this limit, the approximate contribution of the nonlinear term is either $\log\left( \frac{P(G=g_0|C=c_0)}{P(G=g_1|C=c_0)} \right)$, or $\log\left( \frac{P(G=g_0|C=c_1)}{P(G=g_1|C=c_1)} \right)$. Finally, in the limit of large thresholds, or equivalently, large decision times $|z_c^\tau + z_0^c|$ will be a large, $e^{-|z_c^\tau + z_c^0|}$ will be small, and the nonlinear term in (11) can be approximated by a linear function of $z_c^\tau + z_0^c$ obtained using the first order Taylor series expansion. In all these cases, (11) can be approximated by a sum of two SPRTs. However, this approximation may not hold

in general and we suspect many interesting cases will require us to consider the nonlinear model in (11). In those cases, the signal and noise characteristics of context and target will have different – and we think distinguishable – effects on the RT distributions we measure.

## 5 The internal-context update and application to a new task

Recall our promise to explore two extremes on the dimension of context and onset timing, and two extremes on the dimension of context-response dependence. The flanker task is an external context task with a context-independent response, so we now turn to an internal context task with context-dependent response. This task is the AX Continuous Performance Test (AX-CPT), a task with origins in the psychiatry literature now applied to cognitive control [6]. In this task, subjects are asked to make a response to a probe (target) stimulus, by convention labeled 'X' or 'Y', where the response mapping is determined by a previously seen cue (context) stimulus, 'A' or 'B'. In our notation: $g_0 = X, g_1 = Y, c_0 = A, c_1 = B$. Unlike the flanker, where all stimuli pairs are equally likely, in the AX-CPT AX trials are usually the most common (appearing 50% of the time or more), and BY trials least common. AY and BX trials appear with equal frequency, but have dramatically different conditional probabilities due to the preponderance of AX trials.

Two response mappings are used in the literature: an asymmetric one where one response is made on AX trials and the other response otherwise; and a symmetric variant where one response is made to AX and BY trials, and the other to AY and BX trials. We focus on the symmetric variant, since in this case the response is always context-dependent (in the asymmetric variant the response is is context-independent on Y trials). We can use the definition of the task to write a new form for $r(\cdot)$:

$$r(P(C,G)) = \begin{cases} r_0 = \text{'left'} & \text{with probability } P(G = g_0, C = c_0) + P(G = g_1, C = c_1) \\ r_1 = \text{'right'} & \text{with probability } P(G = g_0, C = c_1) + P(G = g_1, C = c_0) \end{cases} \tag{12}$$

We assume for simplicity that the inference process on the context models the maintenance of context information and retrieval of the response rule (though the model could be extended to perceptual encoding of the context as well). That is, we start the inference machine at $t_c^{off}$, using the following update when $t_c^{off} \le \tau \le t_g^{on}$:

$$P_\tau(C, G \mid e^C) \propto P(e^C \mid C, G) P_{\tau-1}(C, G) \tag{13}$$

Then, once the target appears the update becomes:

$$P_\tau(C, G \mid e^C, e^G) \propto P(e^C, e^G \mid C, G) P_{\tau-1}(C, G) \tag{14}$$

For samples after the context disappears, we introduce a simple decay mechanism wherein the probability with which the context sensor provides a sample from the true context decays exponentially. A sample is drawn from the true context with probability $e^{-d(\tau-t_c^{off})}$, and drawn uniformly otherwise. The update takes this into account, such that as $\tau$ grows the ratio $\frac{P(e^C|C=c_0)}{P(e^C|C=c_1)}$ approaches 1 and the context sensor stops being informative (notation omitted for space). This means that the unconditional posterior of context can saturate at values other than 1. The remainder of the model is exactly as described above. This provides an opportunity to generate predictions of both tasks in a shared model, something we take up in the final portion of the paper. But first, as in the flanker model, we reduce this model to a combination of multiple instances of the well-understood DDM.

## 6 Relating the internal context model to the fixed-context drift model

We sketch an intuition for how our internal context model can be built up from a combination of fixed-context drift models (again assuming sensor independence). The derivation uses the same trick of dividing numerator and denominator by the likelihood as the flanker expressions, and is included in the supplementary material, as is the asymmetric variant. We state the final expression for the symmetric case here:

$$\log Z = \log \frac{P_0(C = c_0, G = g_0)e^{z_c^\tau}e^{z_g^\tau} + P_0(C = c_1, G = g_1)}{P_0(C = c_0, G = g_1)e^{z_c^\tau} + P_0(C = c_1, G = g_0)e^{z_g^\tau}} \tag{15}$$

Equation (15) combines the SPRT statistic associated with the context and the target in a nonlinear fashion which is more complicated than in (11), further complicated by the fact that the memory decay turns the context random walk into an Ornstein-Uhlenbeck process in expectation (notation omitted for space, but follows from the relationship between continuous O-U and discrete AR(1) processes). The reduction of these equations to a SPRT or the sum of two SPRTs is subtle, and is valid only in rather contrived settings. For example, if the drift rate for the target is much higher

than the drift rate for the context, then in the limit of large thresholds (15) can be approximated by either $\log \frac{P_0(C=c_0,G=g_0)}{P_0(C=c_1,G=g_0)} + z_c^\tau$, or $\log \frac{P_0(C=c_1,G=g_1)}{P_0(C=c_0,G=g_1)} - z_c^\tau$. As with (11), we think it will be highly instructive to further invesgate the cases where the reductions cannot apply.

# 7  Simulation results for both tasks using the same model and parameters

With the relationship between both tasks established via our theory, we can now simulate behavior in both tasks under nearly the same model parameters. The one difference is in the memory component, governed by the memory decay parameter $d$ and the target onset time $\tau_{t^{on}}$. Longer intervals between context disappearance and target appearance have the same effect as higher values of $d$: they make context retrieved more poorly. We use $d = 0.0001$ for the decay and a 2000-timestep interval, which results in approximately 82% probability of drawing a correct sample by the time the target comes on. The effect of both parameters is equivalent in the results we show, since we do not explore variable context-target delays, but could be explored by varying this duration.

For simplicity we assume the sampling distribution for $e^C$ and $e^G$ is identical for both tasks, though this need not hold except for identical stimuli sampled from perception. For flanker simulations we use the model no spatial uncertainty, i.e. $\alpha_\mu = \alpha_\sigma = 0$, to best match the AX-CPT model and our analytical connections to the SPRT. We assume the model has a high congruence prior for the flanker model, and the correct prior for the AX-CPT, as detailed in Table 1.

| Context | | Target | | Prior | |
|---------|--------|---------|--------|---------|--------|
| Flanker | AX-CPT | Flanker | AX-CPT | Flanker | AX-CPT |
| S_S | A | S | X | 0.45 | 0.5 |
| S_S | A | H | Y | 0.05 | 0.2 |
| H_H | B | S | X | 0.05 | 0.2 |
| H_H | B | H | Y | 0.45 | 0.1 |

Table 1: Priors for the inference process for the Flanker and AX-CPT instantiation of our theory.

The remainder of parameters are identical across both task simulations: $\sigma_c = \sigma_g = 9, \theta = 0.9$, $\mu_c = \mu_g = 0$ for $c_0$ and $g_0$, and $\mu_c = \mu_g = 1$ for $c_1$ and $g_1$. To replicate the flanker results, we followed [5] by introducing a non-decision error parameter $\gamma = 0.03$: this is the probability of making a random response immediately at the first timestep. We simulated 100,000 trials for each model. Figure 1 shows results from the simulation of the flanker task, recovering the characteristic early below-chance performance in incongruent trials. This simulation supports the assertion that our theory generalizes the flanker model of [5], though we are not sure why our scale on timesteps appears different by about 5x in spite of using what we think are equivalent parameters. A library for simulating tasks that fit in our framework and code for generating all simulation figures in this paper can be found at `https://github.com/mshvartsman/cddm`.

For the AX-CPT behavior, we compare qualitative patterns from our model to a heterogeneous dataset of humans performing this task (n=59) across 4 different manipulations with 200 trials per subject [24]. The manipulations were different variants of "proactive"-behavior inducing manipulations in the sense of [25]. This is the most apt comparison to our model: proactive strategies are argued to involve response preparation of the sort that our model reflects in its accumulation over the context before the target appears.

Figure 2 shows mean RTs and accuracies produced by our model for the AX-CPT, under the same parameters that we used for the flanker model. This model recovers the qualitative pattern of behavior seen in human subjects in this task, both RT and error proportion by condition showing the same pattern. Moreover, if we examine the conditional RT plot (Figure 3) we see that the model predicts a region of below-chance performance early in AY trials but not other trials. This effect appears isomorphic to the early congruence effect in the flanker task, in the sense that both are caused by a strong prior biased away from the correct response: on incongruent trials given a high congruence prior in the flanker, and on AY trials given a high AX prior in AX-CPT. More generally, the model recovers conditional accuracy curves that look very similar to those in the human data.

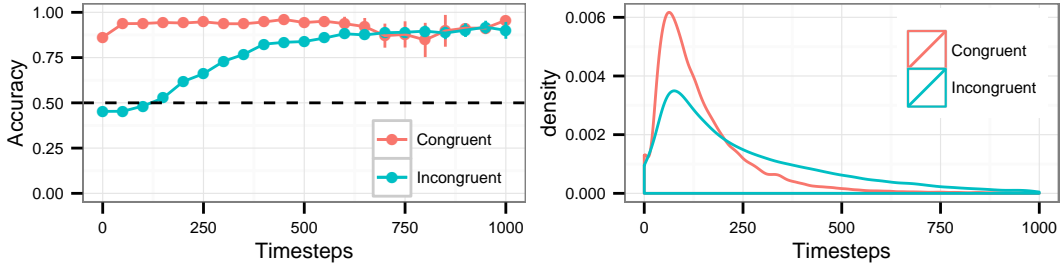

Figure 1: **Model recovers characteristic flanker pattern.** *Left*: response time computed by 50-timestep RT bin for congruent and incongruent trials, showing early below-chance performance. *Right*: response time distributions for congruent and incongruent trials, showing the same mode but fatter tail for incongruent relative to congruent trials. Both are signature phenomena in the flanker task previously recovered by the model of Yu and colleagues, consistent with our theory being a generalization of their model.

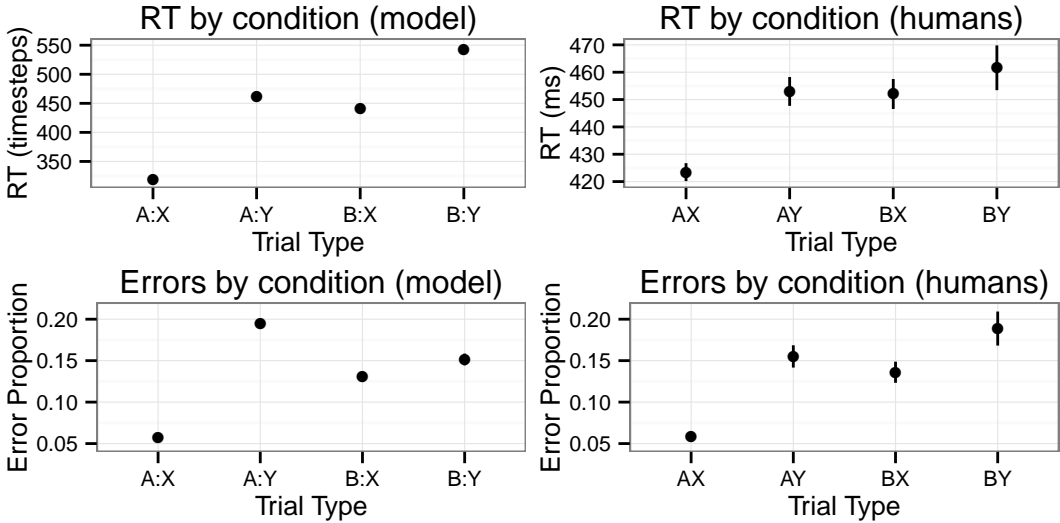

Figure 2: **Model recovers gross RT patterns in human behavior.** *Left*: RT and error rates by trial type in the model, using the same parameters as the flanker model. *Right*: RT and error rates by trial type in 59 human participants. Error bars are standard errors (where not visible, they are smaller than the dots). Both RT and error patterns are quite similar (note that the timestep-to-ms mapping need not be one-to-one).

## 8   Discussion

In this paper, we have provided a theoretical framework for understanding decision making under dynamically shifting context. We used this framework to derive models of two distinct tasks from the cognitive control literature, one a notational equivalent of a previous model and the other a novel model of a well-established task. We showed how we can write these models in terms of combinations of constant-drift random walks. Most importantly, we showed how two models derived from our theoretical framing can recover RT, error, and RT-conditional accuracy patterns seen in human data without a change of parameters between tasks and task models. Our results are quantitatively robust to small changes in the prior because equations 12 and 16 are smooth functions of the prior. The early incongruent errors in flanker are also robust to larger changes, as long as the congruence prior is above 0.5. The ordering of RTs and error rates for AX-CPT rely on assuming that participants at least learn the correct ordering of trial frequencies – we think an uncontroversial assumption.

One natural next step should be to generate direct quantitative predictions of behavior in one task based on a model trained on another task – ideally on an individual subject level, and in a task

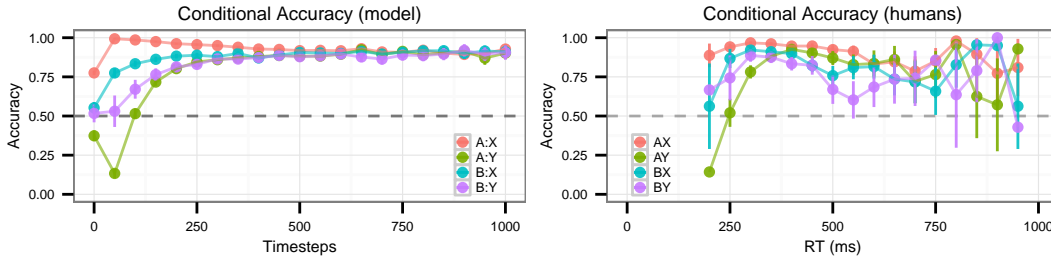

Figure 3: **Model recovers conditional accuracy pattern in human behavior.** *Left*: response time computed by 50-timestep bin for the four trial types, using same parameters as the flanker model. *Right*: same plot from 59 human participants (see text for details). Bins with fewer than 50 observations omitted. Error bars are standard errors (where not visible, they are smaller than the dots). Both plots show qualitatively similar patterns. Two discrepancies are of note: first, the model predicts very early AY responses to be more accurate than slightly later responses, and early B responses to be close to chance. We think at least part of this is due to the non-decision error $\gamma$, but we retained it for consistency with the flanker model. Second, the humans show slightly better BY than BX performance early on, something the model does not recover. We think this may have to do with a global left-response bias that the model is somehow not capturing. Note: the abscissae are in different units (though they correspond surprisingly well).

that fits in our framework that has not been extensively explored (for example, an internal-context Flanker variant, or a context-dependent response congruence judgment task). The main challenge in pursuing this kind of analysis is our ability to efficiently estimate and explore these models which, unlike the fixed-context models, have no closed-form analytic expressions or fast approximations. We believe that approximations such as those provided for the flanker model [22] can and should be applied within our framework, both as a way to generate more efficient data fits, and as a way to apply the tools of dynamical systems analysis to the overall behavior of a system. Of particular interest is whether some points in the task space defined in our framework map onto existing descriptive decision models [e.g. 3].

Another natural next step is to seek evidence of our proposed form of integrator in neural data, or investigate plausible neural implementations or approximations to it. One way of doing so is computing time-varying tuning curves of neural populations in different regions to the individual components of the accumulators we propose in equations (11) and (15). Another is to find connectivity patterns that perform the log-sum computation we hypothesize happens in the integrator. A third is to look for components correlated with them in EEG data. All of these methods have some promise, as they have been successfully applied to the fixed context model [9, 10, 26]. Such neural data would not only test a prediction of our theory, but also – via the brain locations found to be correlated – address questions we presently do not address, such as whether the dynamic weighting happens at the sampler or further upstream (i.e. whether unreliable evidence is gated at the sampler or discounted at the integrator).

A second key challenge given our focus on optimal inference is the fact that the fixed threshold decision rule we use is suboptimal for the case of non identically distributed observations. While the likelihoods of context and target are independent in our simulations, the likelihoods of the two responses are not identically distributed. The optimal threshold is generally time-varying for this case [27], though the specific form is not known.

Finally, while our model recovers RT-conditional accuracies and stimulus-conditional RT and accuracy patterns, it fails to recover the correct pattern of accuracy-conditional RTs. That is, it predicts much faster errors than corrects on average. Future work will need to investigate whether this is caused by qualitative or quantitative aspects of the theoretical framework and model.

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
