[Supplementary Material]

# 1 Appendix: Notational equivalence to the Yu and colleagues flanker model

Yu et al. use the following notation for their update:

$$P(s_2, M \mid X_t) = \frac{p(s_t \mid s_2, M)p(s_2, M \mid \mathbf{X}_{t-1})}{\sum_{s_2', M} p(s_t' \mid s_2, M)p(s_2', M \mid \mathbf{X}_{t-1})} \tag{1}$$

In their notation, the stimulus array is indexed such that $s_2$ is the target and $s_{1,3}$ are the flankers. Therefore, their $s_2$ is simply our $G$. Their $M$ is a trial compatibility or congruence variable, taking on the values of I(ncongruent) and C(ongruent). This gives a straightforward remapping from their joint probability space over target identity and congruence into our space of context and target:

| Stimulus | C | G == $s_2$ | M |
|---|---|---|---|
| SSS | S_S | S | Congruent |
| HHH | H_H | H | Congruent |
| SHS | S_S | H | Incongruent |
| HSH | H_H | S | Incongruent |

Their prior $P(s_2, M \mid \mathbf{X}_{t-1})$ is equivalent to our prior (it is simply the posterior from the previous timestep). Their input $x_t$ is an input vector concatenating the input vectors from the target and two flankers, $[x_1, x_2, x_3]$, such that:

$$x_1(t) \sim (N)(\alpha_1\mu_1 + \alpha_2\mu_2, \sigma_1^2 + \sigma_2^2) \tag{2}$$
$$x_2(t) \sim (N)(\alpha_1\mu_2 + \alpha_2\mu_1 + \alpha_2\mu_3, \sigma_1^2 + 2\sigma_2^2) \tag{3}$$
$$x_3(t) \sim (N)(\alpha_1\mu_3 + \alpha_2\mu_2, \sigma_1^2 + \sigma_2^2) \tag{4}$$

Since the two flanker stimuli are always identical in this experiment, we can define $\mu_c := \mu_1 = \mu_3$ and $\mu_g := \mu_2$. Next, we divide the means by $\alpha_1$, and map $\frac{\alpha_2}{\alpha_1} := \alpha_m u$ to make $x_2(t)$ equivalent to $e^G$. Since the three likelihoods are multiplied and the two flanker likelihoods are identical, updating jointly on $[x_1, x_3]$ will be equivalent to updating twice on two draws of $e^C$.

Yu and colleagues also summarize their prior by defining $\beta$ to be the prior probability of a congruent trial. We can define the priors in the following way to reflect this:

$$P_0(C = c_0, G = g_0) = \frac{\beta}{2} \tag{5}$$

$$P_0(C = c_0, G = g_1) = \frac{1 - \beta}{2} \tag{6}$$

$$P_0(C = c_1, G = g_0) = \frac{1 - \beta}{2} \tag{7}$$

$$P_0(C = c_1, G = g_1) = \frac{\beta}{2} \tag{8}$$

# 2 Full derivation of AX-CPT log likelihood expressions

In the internal context AX-CPT, $t_{c^{on}} \neq t_{g^{on}}$, so we index context samples using $\ell$ and target samples using $t$. We therefore define $l_t(t_x) = P(e_t^G \mid G = g_x)$ and $l_\ell(c_x) = P(e_\ell^C \mid C = c_x)$ for the likelihoods, with $x \in \{0, 1\}$ indexing stimuli. We can write the log likelihood for the two responses to the symmetric AX-CPT, divide numerator and denominator by the product of $g_1$ and $c_1$ likelihoods, and then rewrite the log likelihood ratios into the $z$ term that evolve as biased Wiener processes in the continuum limit. Note that here the context and target walks start at different times.

$$\log Z = \log \frac{P_0(C = c_0, G = g_0) \prod_{\ell=t_c^{on}}^{\tau} l_\ell(c_0) \prod_{t=t_g^{on}}^{\tau} l_t(g_0) + P_0(C = c_1, G = g_1) \prod_{\ell=t_c^{on}}^{\tau} l_\ell(c_1) \prod_{t=t_g^{on}}^{\tau} l_t(g_1)}{P_0(C = c_0, G = g_1) \prod_{\ell=t_c^{on}}^{\tau} l_\ell(c_0) \prod_{t=t_g^{on}}^{\tau} l_t(g_1) + P_0(C = c_1, G = g_0) \prod_{\ell=t_c^{on}}^{\tau} l_\ell(c_1) \prod_{t=t_g^{on}}^{\tau} l_t(g_0)} \tag{9}$$

$$= \log \frac{P_0(C = c_0, G = g_0) \prod_{\ell=t_c^{on}}^{\tau} \frac{l_\ell(c_0)}{l_\ell(c_1)} \prod_{t=t_g^{on}}^{\tau} \frac{l_t(g_0)}{l_t(g_1)} + P_0(C = c_1, G = g_1)}{P_0(C = c_0, G = g_1) \prod_{\ell=t_c^{on}}^{\tau} \frac{l_\ell(c_0)}{l_\ell(c_1)} + P_0(C = c_1, G = g_0) \prod_{t=t_g^{on}}^{\tau} \frac{l_t(g_0)}{l_t(g_1)}} \tag{10}$$

$$= \log \frac{P_0(C = c_0, G = g_0) \sum_{\ell=t_c^{on}}^{\tau} \log \frac{l_\ell(c_0)}{l_\ell(c_1)} \sum_{t=t_g^{on}}^{\tau} \log \frac{l_t(g_0)}{l_t(g_1)} + P_0(C = c_1, G = g_1)}{P_0(C = c_0, G = g_1) \sum_{\ell=t_c^{on}}^{\tau} \log \frac{l_\ell(c_0)}{l_\ell(c_1)} + P_0(C = c_1, G = g_0) \sum_{t=t_g^{on}}^{\tau} \log \frac{l_t(g_0)}{l_t(g_1)}} \tag{11}$$

$$= \log \frac{P_0(C = c_0, G = g_0) e^{z_c^\tau} e^{z_g^\tau} + P_0(C = c_1, G = g_1)}{P_0(C = c_0, G = g_1) e^{z_c^\tau} + P_0(C = c_1, G = g_0) e^{z_g^\tau}} \tag{12}$$

We can do the same for the asymmetric variant.

$$\log Z = \log \frac{P_0(C = c_0, G = g_0) \prod_{\ell=t_c^{on}}^{\tau} l_\ell(c_0) \prod_{t=t_g^{on}}^{\tau} l_t(g_0)}{\begin{aligned} &P_0(C = c_0, G = g_1) \prod_{\ell=t_c^{on}}^{\tau} l_\ell(c_0) \prod_{t=t_g^{on}}^{\tau} l_t(g_1) + P_0(C = c_1, G = g_0) \prod_{\ell=t_c^{on}}^{\tau} l_\ell(c_1) \prod_{t=t_g^{on}}^{\tau} l_t(g_0) \\ &+ P_0(C = c_1, G = g_1) \prod_{\ell=t_c^{on}}^{\tau} l_\ell(c_1) \prod_{t=t_g^{on}}^{\tau} l_t(g_1) \end{aligned}} \tag{13}$$

$$= \log \frac{P_0(C = c_0, G = g_0) \prod_{\ell=t_c^{on}}^{\tau} \frac{l_\ell(c_0)}{l_\ell(c_1)} \prod_{t=t_g^{on}}^{\tau} \frac{l_t(g_0)}{l_t(g_1)}}{P_0(C = c_0, G = g_1) \prod_{\ell=t_c^{on}}^{\tau} \frac{l_\ell(c_0)}{l_\ell(c_1)} + P_0(C = c_1, G = g_0) \prod_{t=t_g^{on}}^{\tau} \frac{l_t(g_0)}{l_t(g_1)} + P_0(C = c_1, G = g_1)} \tag{14}$$

$$= \log \frac{P_0(C = c_0, G = g_0) \sum_{\ell=t_c^{on}}^{\tau} \log \frac{l_\ell(c_0)}{l_\ell(c_1)} \sum_{t=t_g^{on}}^{\tau} \log \frac{l_t(g_0)}{l_t(g_1)}}{P_0(C = c_0, G = g_1) \sum_{\ell=t_c^{on}}^{\tau} \log \frac{l_\ell(c_0)}{l_\ell(c_1)} + P_0(C = c_1, G = g_0) \sum_{t=t_g^{on}}^{\tau} \log \frac{l_t(g_0)}{l_t(g_1)} + P_0(C = c_1, G = g_1)} \tag{15}$$

$$= \log \frac{P_0(C = c_0, G = g_0) e^{z_c^\tau} e^{z_g^\tau}}{P_0(C = c_0, G = g_1) e^{z_c^\tau} + P_0(C = c_1, G = g_0) e^{z_g^\tau} + P_0(C = c_1, G = g_1)} \tag{16}$$