[Reviews · NeurIPS 2015]

Submitted by Assigned_Reviewer_1

While the overall performance of the model doesn't seem especially impressive given the small number of phenomena being considered and the apparent number of potential moving parts in the model, it may nonetheless be worth publishing. Diffusion models are not my area of expertise and this was a "light" review, hence my low confidence score.

[Edit: Having read the other reviewers' comments and the author's response, I've revised my rating a point upward.]
Summary: The paper proposes an extension of an existing diffusion model of decision-making to cover a broader range of rules than previous approaches. The generality of the new model is nice.

Submitted by Assigned_Reviewer_2

The authors extend the popular sequential probability ration test for decisions under perceptual uncertainty to situations in which a there are additional contextual stimuli. This is achieved by simply accumulating samples and computing a posterior distribution over both context and target from which the decision is being generated when a bound is being reached. As the onset and offset of the cues can be modeled separately, the authors decided to give examples for the cases in which target and context appear at the same time and where the target cue appears with a delay after the disappearance of the contextual cue, corresponding to the Eriksen flanker task and the continuous performance test.

Finally, the authors set the parameters for the flanker task and use these same parameters for the second task. The behavioral performance is matched well and the corresponding reaction times are within 40% of the empirically measured values of the experiment.

This is overall an interesting topic for the cognitive science and neuroscience community that follows NIPS and it is an extension of the sequential probability ration test, which has been extremely popular in the last years. On the other hand, there is other recent work emphasizing the importance of modeling contextual effects in explain human decision making behavior: the authors should relate their approach to previous work in decision-making with contexts e.g. Srivastava & Schrater's work, at least cite it. The current model assumes a perfect separation of the evidence provided for target and context given the sensory stimuli, which is potentially not very realistic. I would call the proposed model a 'model' and not a 'theory' because a model can certainly generate or accommodate multiple experimental settings, but this may be a matter of taste.
Summary: Extension to the popular random walk to bound decision under perceptual uncertainty model to include contextual stimuli. Data of two experiments can be fit within one framework and with the same set of parameters.

Submitted by Assigned_Reviewer_3

This paper uses a Bayesian formulation of sequential decision making to investigate the effect of context. Context is defined as additional information becoming available either before or at the moment of the stimulus onset. Simulation results are compared to earlier human data across different tasks without changing the model parameters.

I am not sure about the overall message of this paper. Even though the advertised goal is to provide a theoretical framework for the effect of context, it appears as if the authors show the effect of sequentially arriving information in an inference making process. It can be called "context"' but -as the authors also point it out- this is just time difference. But if it is so, then are we talking about a special effect of context or just exploring the dynamics of inference making?

The sample tasks are fairly simple, e. g. the patterns of the congruent/incongruent task are mostly expected and the one which is more surprising (below chance performance at the beginning) is not explained. Yu et al. talks about attention, but there is no attention mentioned in the present paper - so, again, is this effect attention-related or a consequence of the inference making process? The transfer from the flanker test to the AX-CPT test is not convincing, i.e. the model and human data is only roughly similar and it is not clear how good the similarity should be so that it could not be replicated by some slightly adjusted random model.

The quantitative predictions that the authors mention in the discussion as a "natural next step" would have been nice.

Summary: An unclear presentation of derivation and results about sequential information processing in a probabilistic inference framework.

Submitted by Assigned_Reviewer_4

This paper presents a unified account of a variety of tasks using a probabilistic theory of context-dependent decision making. In essence, it is an extension of classical signal detection theory. While other papers have analyzed special cases of this framework, this paper not only connects them together but demonstrates that performance on different tasks can be accounted for using the same model parameters. Overall, I was impressed.

Major comments:

- p. 2: the notation in section 2 is confusing. Initially C and G are referred to as distributions, but seem to be treated as sets. Then they are treated as random variables with distribution P(C,G). The latter seems to be the correct treatment.

- r() is defined as a mapping from P(C,G) onto a response probability, but in Eq. 3 it is defined as a function of the random variables themselves. Since the decision maker does not have direct access to these variables, this doesn't seem correct. The marginals used in the decision rule should be explicitly described.

- No mention is made of earlier work that modeled internal context updating, notably O'Reilly & Frank (2006) and Todd, Niv & Cohen (2009). These models were framed in terms of reinforcement learning, rather than perceptual decision making. I think it would be valuable to comment on the strengths and weaknesses of these approaches, and how they might be combined.

Minor comments:

- p. 1: "looks a friend" -> "is looking for a friend"

- throughout, the equation font is very small.

- p. 6: "moreso" -> "more so"

- p. 6: "result patterns" -> "resulting patterns"
Summary: A valuable synthesis and extension of previous decision making models.

Author Feedback
Author rebuttal: We recognize that we overemphasized the reduction to the Yu et al. flanker model relative to of our main contribution: generalizing the theory to a broader range of task conditions, while preserving the flanker model as a special case. In particular, we showed how the framework behaves when sources of information are not presented simultaneously, and involve qualitatively different patterns of stimulus-response dependencies. We will adjust this emphasis in our presentation of the work. We next address individual reviewers:

Reviewer 1: we agree that Srivastava & Scharter's work bears mentioning. However, they update once for each decision, addressing only between-trial adjustments. Our focus is on within-trial dynamics and response times, a key departure from their work. At the same time, they consider many contexts whereas we tackle one, so the approaches are complementary. On independence: the theory does not in general require independent evidence (see eqns. 4, 5), but we do require it for the analytical results in Sec. 4, and use it for easier simulation in Sec. 7.

Reviewer 3: we agree that we overloaded notation in ways we did not make sufficiently clear. What we mean is: C and G are random variables; c and g are trial-level unobserved draws from C and G, taking on possible values in sets curly-C and curly-G; e^C and e^G are within-trial evidence draws from which c and g are inferred. r(.) always maps from the posterior over a single trial's c and g to a response. This mapping marginalizes over the value of c for the flanker task but does not for AX-CPT. We will clarify this in revision. The RL models mentioned are valuable and complementary, since (as noted above) our theory explains within-trial rather than between-trial dynamics. We think a natural place to combine both is in using our model in place of decision rules like softmax, which typically do not use response times to understand the learning process.

Reviewer 4: the reviewer expressed concern about our operationalization of one source of information as "context." We agree that our model can be used to model any two sources of information, but focus on tasks in which one is used as context for processing another. Such situations abound in daily life and have been the focus of extensive research, including in the tasks we model (e.g. Eriksen & Eriksen 1974; Cohen & Servan-Schreiber 1992), so we think viewing them together in a single theory is a useful contribution. However, we recognize that the model may have wider applicability, and we will emphasize this in revision.
With regard to attention, this is often interpreted in models like ours as influencing the rate of evidence accumulation (e.g. White, Ratcliff & Starns 2011). Our model, as a generalization of the Yu et al. model, offers a normative account of the time-varying relative weighting of two sources of evidence (e.g., the target and flankers, or AX-CPT cue and probe). One can therefore view our model as defining the optimal attentional policy over two evidence sources.
Regarding fits: we think that generating the correct scales and orderings of data in two tasks using a single unfit parameter set is nontrivial, especially in spite of their differences mentioned above, and supports the generality of the model. We discuss below how the patterns we recover should be robust to parameter choice. With that said, we share the reviewer's concern over direct fit as a more stringent test of our theoretical claims and a better descriptive account of data. Better direct fits are possible, and might be aided by replacing our heuristic fixed threshold with a time-varying optimal threshold. More practically, we imagine that tools from the diffusion model literature (e.g. start point variability, non-decision times) could be used to yield improved descriptive fits, and will mention this in the revision.

Reviewer 5: we appreciate recognition of the usefulness of our approach. We hope that concerns about clarity were addressed in the responses to other reviewers.

Reviewer 6: the conditional probabilities are computed from Table 1. Our results are robust to small changes in the prior because eqns. 12 and 16 are smooth functions of the prior. The early incongruent errors in the flanker task are robust to larger changes as long as the congruence prior is above 0.5, exactly as in Yu et al. The ordering of RTs and error rates for AX-CPT trial types rely on assuming that subjects (at minimum) learn the correct ordering of trial frequencies.

Reviewer 7: we think the number of moving parts in our model is small, especially in light of producing accuracy-conditional RT patterns in multiple trial types in multiple tasks by using a single, pre-determined parameter setting. Existing models of flanker have at least the same number of parameters we use (e.g. White, Ratcliff & Starns 2011), and these other models do not extend to the AX-CPT with a single additional parameter.